# Causes of Pulmonary Fibrosis in the Elderly

**DOI:** 10.3390/medsci6030058

**Published:** 2018-07-24

**Authors:** Cecilia López-Ramírez, Lionel Suarez Valdivia, Jose Antonio Rodríguez Portal

**Affiliations:** 1Unidad Médico-Quirúrgica de Enfermedades Respiratorias, Hospital Universitario Virgen del Rocío, Instituto de Biomedicina de Sevilla, 41013 Sevilla, Spain; ceclopram@gmail.com (C.L.-R.); lionel_278@hotmail.com (L.S.V.); 2Centro de Investigación en Red de Enfermedades Respiratorias (CIBERES), Instituto de Salud Carlos III, 28029 Madrid, Spain

**Keywords:** aging, idiopathic pulmonary fibrosis, extracellular matrix, fibroblasts, alveolar epithelial cell

## Abstract

Idiopathic pulmonary fibrosis (IPF) is the most common and most lethal type of idiopathic interstitial pneumonia. It is a chronic, aging-associated lung disease characterized by fibrotic foci and inflammatory infiltrates, with no cure and very limited therapeutic options. Although its etiology is unknown, several pathogenic pathways have been described that could explain this process, involving aging, environmental factors, genomic instability, loss of proteostasis, telomere attrition, epigenetic changes, mitochondrial dysfunction, cell senescence, and altered intercellular communication. One of the main prognostic factors for the development of IPF in broad epidemiological studies is age. The incidence increases with age, making this a disease that predominantly affects the elderly population, being exceptional under 45 years of age. However, the degree to which each of these mechanisms is involved in the etiology of the uncontrolled fibrogenesis that defines IPF is still unknown. Clarifying these questions is crucial to the development of points of intervention in the pathogenesis of the disease. This review briefly summarizes what is known about each possible etiological factor, and the questions that most urgently need to be addressed.

## 1. Introduction

Idiopathic pulmonary fibrosis (IPF) is a chronic progressive lung disease usually affecting the elderly, and its appearance is exceptional below the age of 45 years [1]. Its incidence and prevalence increase with age, and most cases are diagnosed in patients older than 60 years [2]. The pathophysiology of the disease is characterized by a fibroproliferative disorder involving excessive deposition of collagen and elements of the extracellular matrix in the lung parenchyma as a consequence of the proliferation and activation of fibroblasts [3]. However, the causes that lead to the uncontrolled fibrogenesis characteristic of the disease are still unknown, and it is difficult to identify the possible risk factors, given that patients are typically diagnosed at an advanced stage of the disease. In fact, epidemiological studies have shown that one of the main predictors of IPF diagnosis is indeed age.

Aging is a complex multifactorial process characterized by a progressive loss of physiological integrity, accumulating deleterious changes of tissues, and cell damage, which are responsible for increased vulnerability and risk of diseases and death [4]. Aging is also associated with increased susceptibility to a wide range of chronic diseases, including lung diseases such as IPF. Our understanding of the biology of aging has advanced significantly in recent years. We know that even in the absence of disease, both structural and functional age-related changes appear in the lungs. It has been suggested that IPF is a consequence of accelerated aging of the lungs, or that pathogenic mechanisms develop there more easily after a certain age. It has been postulated that in genetically predisposed subjects, diverse exogenous factors such as environmental and occupational exposure or infection attack the alveolar epithelium, generating the fibrotic process [5]. However, how and to what extent they participate in the development of IPF is uncertain.

## 2. Impact of Aging on Respiratory System

Physiologic aging of the lungs is associated with anatomical and functional changes, including decreased elastic recoil in both alveoli and airways, a progressive decrease in compliance of the chest wall, and a decrease in the strength of respiratory muscles. These changes are caused by dilation of alveoli, increases in the size of airspaces, a decrease in exchange surface area, and loss of supporting tissue for small airways. The vital capacity decreases and the residual volume increases. Expiratory flows decrease with a characteristic alteration in the flow-volume curve, suggesting small airway dysfunction. The ventilation-perfusion heterogeneity increases and carbon monoxide transfer decreases with age as a result of loss of airway, reduction of alveolar surface area and alteration of the alveolus-capillary membrane [6].

Epidemiological studies indicate that lung aging is associated with higher susceptibility to common chronic respiratory diseases such as chronic obstructive pulmonary disease (COPD) and IPF. Although COPD and IPF are distinct disease entities, they share some similarities. Both increase with age, both are punctuated by episodes of exacerbations, and histopathologic studies in animal models have shown increased collagen deposition and progressive pulmonary fibrosis in the lungs of aging rats [7], (although, admittedly, this occurs in different locations in each disease, in the small airways in patients with COPD and in the lung parenchyma in IPF) [8]. In addition, both are associated with chronic structural remodeling of the extracellular matrix and impaired pulmonary function, and interestingly both conditions can coexist in the same patient [9].

Radiographic studies have found alterations suggestive of interstitial disease when high resolution computed tomography was performed for unrelated reasons in asymptomatic elderly individuals with normal pulmonary function. However, it is not clear whether these changes are a consequence of lung aging or represent clinically relevant interstitial disease [10]. Other imaging and animal model studies have shown that the aged lung is usually the so-called “senile” emphysema, characterized by distal airspace enlargement with progressive loss of elastic recoil [11]. Moreover, prematurely aged lungs develop emphysematous changes rather than fibrosis. Because of the great variety of cell types in the lung, the consequences of aging display a broad spectrum of phenotypes. Clarifying the mechanisms responsible for the pathogenesis of fibrosis and disrepair associated with aging has been the goal of numerous studies.

## 3. Environmental and Endogenous Factors

The development of pulmonary fibrosis with age is at least partly related to the accumulation of environmental factors. These factors include automobile emissions, occupational dust exposure and cigarette smoke [12]. Regardless of their magnitude, these effects of foreign matter inhalation produce alterations in the alveolar epithelial cells (AEC). These cells are a critical component in alveolar homeostasis and proliferate to replace damaged cells, sometimes exacerbating damage that might otherwise be asymptomatic. Aberrantly activated lung epithelial cells produce virtually all the mediators responsible for fibroblast migration, proliferation, and activation with the subsequent exaggerated extracellular matrix accumulation and destruction of the lung parenchyma [13]. The consequences are a decrease in the stem cell differentiation potential, rapidly shortening telomeres, and increases in the probability of mutations involving nucleotide mismatch during DNA replication [14].

Several epidemiological studies support the conclusion that environmental factors have an etiologic role in IPF. Cigarette smoking, especially of more than 20 pack-years, is the most strongly associated environmental risk factor [15]. Recent evidence suggests that emphysema and pulmonary fibrosis are both consequences of smoking, and that they may even coexist in the same susceptible individuals [16]. Recent studies have searched for biomarkers that could help to identify high-risk individuals likely to develop IPF or combined pulmonary fibrosis and emphysema [9]. Other factors such as metal and wood dust exposure have also been associated with an increased risk of IPF. Imbalances in levels of oxidative stress induced by these inhaled agents and disruptions of antioxidant protective mechanisms have been found in patients with IPF [17]. 

In addition to these external factors, endogenous factors also influence the development of IPF. The immune system weakens gradually with age due to internal mechanisms that include T-cell senescence. As a consequence, immunological responses such as reactions to infection or responses to vaccination are impaired. Results of human studies and animal models have shown an association between IPF and Epstein-Barr virus (EBV) infection or gamma-herpes virus infection [18]. Studies in murine models have confirmed that aged wild-type mice experience more severe progressive pulmonary fibrosis than younger mice after gamma herpesvirus infection. In these processes, age plays an important role. Pulmonary fibrosis induced by gamma-herpesvirus in aged mice is associated with increased fibroblast responsiveness to transforming growth factor-beta (TGF-β) [19,20]. In addition, aging mice receiving gamma herpesvirus responded with endoplasmic reticulum (ER) stress, apoptosis of type II lung epithelial cells, and activation of profibrotic pathways [21]. In this study in a murine model, only aged mice (>15 months) developed γ-herpesvirus-68-induced lung fibrosis through a mechanism that involved alveolar epithelial cell reprogramming to produce pro-fibrotic factors and enhanced TGF-β signaling in lung fibroblasts [22]. These associations are further supported by the finding, in the lungs of IPF patients, of both EBV proteins and EBV DNA.

It should also be noted that conditions such as gastroesophageal reflux, pulmonary hypertension, emphysema and radiotherapy may also be risk factors for IPF.

## 4. Genetic and Epigenetic Changes

Some studies suggest that pulmonary fibrosis is genetically determined [23,24]. Aging increases the quantity of mutations while simultaneously decreasing the body’s ability to repair damaged DNA. Emerging genetic studies offer new insights into the fundamental mechanisms of pulmonary fibrosis. The question is how all the genetic alterations identified by the researchers can be incorporated into a unified pathogenic model of IPF.

General genetic mutations and epigenetic methylations accumulating with age release a process that involves excess generation of fibrous tissue and overall damage to the pulmonary system. Mutations in genes of different biologic pathways lead to the common phenotype of familial pulmonary fibrosis (FPF) and sporadic IPF. One of the first genetic associations identified was mutations in the genes encoding the lung surfactant proteins C and A2 (*SFTPC* and *SFTPA2*, respectively) [25]. These proteins are expressed exclusively by type II AECs and are essential for lung function and homeostasis after birth. Mutations in *SFTPC* and *SFTPA2* associated with the disease injure AECs and reduce their repair capacity. These effects are mediated by increased ER stress and changes of the unfolded protein response [26].

Also associated with pulmonary fibrosis are mutations in the genes that maintain the length of the telomeres (*TERT*, *TERC*), which are more common in the familial forms [27]. In contrast to *SFTPC* and *SFTPA2*, telomerase genes are also expressed by extrapulmonary cells, especially stem cells. This raises the possibility that changes in the renovation and repair of AEC after injury predispose to the initiation and progression of the disease. Other genetic associations with IPF have been confirmed by genome-wide association studies (GWAS), including mutations in the promoter region of mucin 5B (MUC5B), a gel-forming protein fund in the secretions of the respiratory tract, and in toll interacting protein (TOLLIP), which participates in the regulation of the immune system and is associated with IPF susceptibility [24]. However, only a small proportion of carriers of these variants in the general population develop IPF, suggesting that genetic predisposition is probably necessary, but not sufficient, to initiate the pathogenic pathways.

Epigenetic mechanisms are also capable of altering gene expression. Epigenetics is the study of heritable changes in the expression of genes that are independent of changes in the DNA sequence [28]. Among the most important epigenetic mechanisms are DNA methylation, histone post-translational modifications and non-coding RNA-mediated gene silencing.

DNA methylation is a major epigenetic mechanism regulating gene expression. DNA methylation refers to the addition of a methyl (CH3) group to the DNA strand itself, on a cytosine or adenine nucleotide, most commonly at a CpG site, catalyzed by DNA methyltransferase. In the elderly there is a progressive loss of DNA methylation in repetitive elements scattered throughout the genome that seems to be proportional to life expectancy [29]. This causes the over-methylation of many promoter regions in the DNA code, which results in the silencing of an increasing number of genes as a person ages. Thus, DNA methylation causes fibroblasts to develop a more aggressive phenotype, with a greater capacity for differentiation into myofibroblasts and greater deposition of collagen matrix [30]. Aberrant DNA methylation can silence or activate gene expression patterns that drive the fibrosis process [31]. Another mechanism that favors the fibrotic process in IPF is changes in histones, for example, reduced histone H3 and H4 acetylation, which are associated with decreased cyclooxygenase-2 (COX-2) expression and prostaglandin E2 (PGE-2—a strong antifibrotic mediator) production by fibroblasts [32]. In a murine model, Sanders et al., 2014 demonstrated that histone deacetylase inhibition promotes fibroblast apoptosis. Suberoylanilide hydroxamic acid (SAHA) induced apoptosis of IPF myofibroblasts, an effect that was mediated, at least in part, by upregulation of the pro-apoptotic gene *Bak* and downregulation of the anti-apoptotic gene *Bcl-xL* [33]. Recent studies implicate the reactive oxygen species (ROS)-generating enzyme NADPH oxidase 4 (Nox4) in cellular senescence [34].

An important type of cell in the development of fibrosis is the myofibroblast. Differentiated myofibroblasts maintain the capacity to proliferate in response to exogenous mitogenic stimuli and are resistant to serum deprivation-induced apoptosis. These proliferative and anti-apoptotic properties of myofibroblasts are related, in part, to the downregulation of caveolin-1 (Cav-1) by TGF-β1 [35]. Alterations in the expression of these apoptosis-related genes are associated with histone modifications and changes in DNA methylation.

MicroRNAs (miRNAs) are posttranscription regulators that bind to specific sequences, blocking translation or causing the degradation of target messenger RNA, resulting in the silencing of genes. miRNAs are expressed in blood and lung tissue of IPF patients and are involved in epithelial repair, epithelial-mesenchymal transition, fibroblast activation, myofibroblast differentiation, macrophage polarization, AEC senescence and collagen production [36]. MiRNAs have been proposed as regulators of cell aging. Altered expression of numerous miRNA types has been linked to the pathogenesis and progression of IPF. Markers of aging, including p16, p21, p53, and beta galactosidase (SA-βgal), were measured in the AECs of lungs with IPF and lungs of healthy donors. The miRNAs were quantified using RT-PCR. Molecular markers of aging (p16, p21 and p53) were elevated in type II pneumocytes with IPF. The activity of SA-βgal was detected in a larger percentage of type II pneumocytes isolated from patients with IPF (23.1%) than of type II pneumocytes isolated from patients with other interstitial diseases (1.2%) or from normal controls (0.8%) [37]. The relative levels of miRNAs associated with aging (miR-34a, miR-34b, and miR-34c, but not miR-20a, miR-29c, or miR-let-7f) were significantly higher in AECs of patients with IPF. For example, the overexpression of miR-34a, miR-34b, and miR-34c in lung epithelial cells was associated with higher levels of SA-βgal activity (27.8%, 35.1% and 38.2% respectively) than the levels observed in untreated control cells (8.8%). Targets of miRNA miR-34, including E2F1, c-Myc, and cyclin E2, were found at lower levels in type II pneumocytes with IPF. These results show that aging markers are elevated only in AEC of IPF and suggest that the miR-34 family of miRNAs regulate aging in AECs [38]. IPF also involves a significant decrease in let-7d miRNA. Let-7 participates in a variety of physiological and pathological mechanisms, in which it plays an important regulatory role. Inhibition of let-7d in vitro induces an epithelial-mesenchymal transition, while inhibition in vivo causes alveolar septal fibrosis [39]. In the same way, in IPF the lungs show upregulation of miR21 and downregulation of miR29, mainly in the myofibroblasts. The increase in miR21 levels promotes fibrogenic activity of TGFβ1 in the fibroblasts, whereas downregulation of miR21 attenuates this activity [40]. In contrast, miR29 levels correlate inversely with the expression of several profibrotic target genes and with the severity of fibrosis. Recently, other miRNAs have been described that could be potential targets for treatment in IPF patients. The novel RNA mimic “miR29b Psh-match” shows a therapeutic effect in bleomycin-induced IPF in model mice [41]. These findings suggest that miRNAs may be diagnostic and therapeutic targets for IPF and also indicators of IPF disease prognosis.

## 5. Loss of Proteostasis

Proteostasis refers to a collection of cellular processes mediating protein folding, misfolding, unfolding, and degradation [42]. Proteostatic stress is handled by two major proteolytic systems, the autophagy-lysosomal system and the ubiquitin-proteosomal system, both of which decline with age. Recent studies have indicated an important role for proteostasis in the development of IPF and other lung diseases.

In IPF, proteostatic mechanisms come into play mainly via their effects on type II AEC. Type II AEC are secretory cells that synthesize and secrete massive quantities of proteins to produce pulmonary surfactants and maintain airway immune defenses. To support this function, type II AEC are equipped with an elaborate ER. In IPF, ER stress facilitates fibrotic remodeling through activation of pro-apoptotic pathways [43]. 

Many researchers consider autophagy to be the main mechanism of protection that cells use to respond to stress states. Autophagy is a catabolic process in which endogenous proteins and damaged organelles are destroyed intracellularly inside autophagosomal membranes, which seal and fuse with lysosomes for degradation of the entrapped cargo [44]. Promotion of autophagy is necessary and sufficient to maintain the normal fate of lung fibroblasts. Studies in vitro have provided evidence that autophagy inhibition accelerates epithelial cell senescence and induces fibroblast-to-myofibroblast differentiation. A recent study using an experimental bleomycin-induced lung fibrosis mouse model deficient in Atg4b (an autophagic protein) found reduced autophagy and a significantly higher inflammatory and fibrotic response. Likewise, we found that Atg4b disruption resulted in augmented apoptosis, affecting predominantly alveolar and bronchiolar epithelial cells [45].

Due to dysfunction of the mechanisms of intracellular autophagy, as cells age various ROS accumulate in their interior. Oxidative stress is an important molecular mechanism underlying fibrosis in a variety of organs, including the lungs [46,47]. Among the major theories of aging is the free radical theory, in which mitochondria are identified as responsible for the initiation of most of the free radical reactions related to the aging process. The increasing age-related oxidative stress is postulated to be the consequence of an imbalance between free radical production and antioxidant defenses with a higher production of the former. However, this concept is overly simplistic for explaining the mechanism of fibrosis at the pulmonary level. 

Initial observations linking ER stress and IPF were made in cases of familial interstitial pneumonia (FIP), the familial form of IPF, in a family with a mutation in surfactant protein C (SFTPC). Subsequent studies involving lung biopsy specimens showed that ER stress was implicated in the pathogenesis of IPF with the discovery of disease-causing mutations in SFTPC, which result in a misfolded gene product in type II AECs [48]. ER stress and the unfolded protein response have been linked to lung fibrosis through regulation of AEC apoptosis, epithelial-mesenchymal transition, myofibroblast differentiation, and M2 macrophage polarization [26]. Although there are advances in knowledge, our understanding of the relationship between ER stress and lung fibrosis is lacking. Burman et al. published recently an interesting review discussing potential causes of ER stress induction in the lungs and current evidence linking ER stress to fibrosis in the context of individual cell types: AECs, fibroblasts, and macrophages [49].

Multiple factors contribute to oxidative stress in the injured lung, which can impair proteostasis. Environmental toxins (e.g., tobacco, occupational exposure) may induce ROS via induction of ER stress, uncoupling of mitochondrial electron transport or enzymatic systems [50], and production of NADPH oxidase (NOX), especially NADPH oxidase-4 (NOX4), in inflammatory and lung target cells. ROS promote apoptosis of airway epithelial cells and elicit the production of cytokines and growth factors such as TGF-β [51], which may play an important role in the differentiation of invasive myofibroblasts and in collagen deposition, and may impair antioxidant defenses. The susceptibility of epithelial cells to apoptosis may be affected by mutations in surfactant, telomerase and mucin genes. 

Proteomic analyses of IPF lung tissue provide evidence for the activation of a DNA damage response. Accumulating evidence implicates p53 in the pathophysiology of pulmonary fibrosis. The expression of p53 and p21 are upregulated in association with chronic DNA damage in IPF lung epithelial cells, mediating either G1 arrest or apoptosis. This shows that deficiencies in DNA repair pathways contribute to the pathogenesis of IPF. 

## 6. Genomic Instability

The accumulation of genetic damage throughout life is a well-recognized component of the aging process. It has been reported that genomic instability, in the form of microsatellite instability and loss of heterozygosity (LOH), occurs in IPF patients. In recent studies, LOH at the homeodomain-interacting protein kinase 2 (*HIPK2*) gene, which is typically activated by genotoxic stimuli, was reported to underlie apoptosis resistance, specifically in IPF fibroblasts [52]. In another study, microsatellite DNA was used to evaluate 52 sputum and venous blood DNA pairs from IPF patients. LOH was found in 20 (38.5%) patients in at least one locus. These alterations were observed in microsatellite DNA markers located in the *MYCL1*, *FHIT*, *SPARC*, p16 (Ink4) and *TP53* genes [53]. 

## 7. Telomere Attrition

Telomeres are ribonucleoprotein structures located at the ends of the chromosomes of eukaryotic cells. This region is a dynamic zone that is highly regulated and consists of a stretch of DNA repeats (TTAGGG) consecutively associated with six protective protein structures called shelterins [54]. The telomeres protect the DNA by preventing the end of the linear chromosome from being recognized as a free end and preventing the ends of the chromosomes from joining together or recombining with other areas of DNA. There is an association between numerous genes in the telomerase maintenance pathway and IPF but the mechanisms by which telomere defects provoke lung disease are not fully understood. There are pathogenic telomerase variants causing dysfunction of telomerase activity leading to accelerated telomere shortening. In the other hand AECs are necessary for repair after pulmonary injury by its activation and proliferation. Every replication of AEC DNA results in the decrease of telomere length. During cell replication, the enzyme DNA polymerase cannot completely replicate the ends of the DNA, so in each cell division a shortening of the telomeres occurs, with eventually fatal consequences for the stability and integrity of the genetic information which result in apoptosis or senescence. However, the enzyme called telomerase adds repeated TTAGGG sequences, which lengthens the ends of the chromosomes and compensates for their shortening during replication. Telomerase is a ribonucleoprotein with polymerase activity, and is formed from two components, a ribonucleotide component that is nothing more than a portion of RNA known as TR (telomerase RNA), and a protein component called TERT (telomerase reverse transcriptase), which reverses the normal course of processing by transcribing RNA to DNA. Thus, the TERT generates and adds to the ends of the chromosomes repeated sequences of TTAGGG, which are formed by the TR portion of the template.

Experimental laboratory models show that inadequate maintenance of telomeres causes cell aging by decreasing cell replication. This may develop in various diseases, including IPF. In lung parenchyma, type II AEC are responsible for repairing alveolar damage by replacing damaged alveolar cells. It has been suggested that cellular aging of these cells and their inability to replicate causes disease at a level that is inversely proportional to the shortening of the telomeres. A recent study supports this hypothesis [55]. In this work, mice were genetically modified to eliminate telomeric repeat-binding factor (Trf2), causing telomere dysfunction that was restricted to type II AEC. The activity of stem cells, as well as their replication in this subpopulation of cells, was affected, which led to aging of the cells, and limited alveolar repair with alterations in the lung parenchyma.

In addition to aging itself and external factors, mutations can inactivate specific genes that have a direct function in the maintenance of the telomeres. There are 11 genes that encode components of telomerase (*TERT*, *NOP10*, *NHP2*, *WRAP53*, *TERC* and *DKC1*) and several proteins (RTEL1, POT1, TPP1, TINF2, CTC1) that form parts of shelterins, the protective structures of telomeres. Mutations in the telomerase genes cause approximately 20% of cases of FPF (defined by two or more individuals of a family with pulmonary fibrosis), and it is characteristic that all these patients have short telomeres. A recent study investigated the association between short telomeres in type II alveolar cells in fibrotic and non-fibrotic areas of patients with IPF, either with or without mutations in the *TERT* gene [56]. It was observed that in the areas of fibrosis of patients with sporadic IPF and mutations in the *TERT* gene, type II AEC had short telomeres, confirming the results of previous studies by Alder et al. [55]. In FPF patients with mutations in the *TERT* gene, however, the telomeres were equally short in fibrotic and non-fibrotic areas. This suggests that patients with telomere mutations are born with lungs that age at an accelerated rate, and are more susceptible to alveolar damage and to development of fibrosis.

Recently, a gene therapy study was carried out. Transgenic mice with mutations in the *Tert* gene were given low bleomycin doses to induce alveolar damage and areas of fibrosis. A *Tert*-carrying adenovirus (AAV9-*Tert*) with high viral transduction at the pulmonary level was used as a vector, increasing the overexpression of TERT mRNA at the level of type II AEC. Eight weeks after the viral inoculation, an increase in lung capacity, a decrease in alveolar collagen content ,and a reduction in the area of fibrosis were found in the mice treated with AAV9-*Tert*; all these in comparison with the control group inoculated with AAV9-vacuum [57]. In summary, these studies demonstrate the possible therapeutic potential of addressing telomeric shortening in pulmonary fibrosis.

## 8. Cellular Senescence

Cell aging or senescence is a process in which cells are stopped at some stage of the cell cycle, preventing their replication and therefore the cellular renewal of damaged tissues. This state of stable replicative arrest occurs with aging and causes imbalances in tissue homeostasis accompanied by secretion of mediators, including pro-inflammatory cytokines and metalloproteinases, collectively termed the “senescence-associated secretory phenotype” (SASP). In IPF pathogenesis, pro-aging stressors have been described that induce premature senescence effects, including telomere attrition, mitochondrial dysfunction, oxidative stress, DNA damage and proteome instability.

Although proliferation arrest is a major hallmark of senescent cells, these cells are not inert; they are metabolically active. Senescent cells may contribute to a loss of tissue homeostasis and impaired organ function through a reduction in the regenerative capacity of tissues, due to proliferative arrest or to the deleterious properties of the SASP [58]. Several factors associated with the SASP, in particular TGF-β and IL-6, increase in the murine lung with age. Removal of senescent cells reduced expression of SASP factors with established roles in regulating fibrotic and pulmonary aspects of IPF, including IL-6, TGF-β and matrix metalloproteinase 12 (MMP12), suggesting that SASP is a major mediator of IPF pathology [59]. These findings suggest that cellular senescence plays a causal role in physiological lung aging and could potentially be targeted to combat age-associated phenotypes in the lung. Treatment of mice with drugs that induce apoptosis specifically in senescent cells, termed senolytics, mirrored the results observed with transgenic cell clearance [60]. Another recent study reported that pulmonary fibrosis, induced in mice following irradiation, was reversed following treatment with ABT-263, a senolytic agent that selectively killed senescent type II alveolar epithelial cells [61].

IPF lung tissue and human IPF cells show increased senescence propensities ex vivo. In an analysis of pulmonary tissues taken from patients with IPF, increases in p16 and p21 were observed in type II AEC that were inversely correlated with carbon monoxide transfer capacity, which suggests that high levels of p16 correlate with an increase in the severity of the disease [62]. Telomere damage may also impact senescence and associated phenotypes via its effects on mitochondria, which are the main generators of ROS within the cell. On the other hand, a study in human MRC5 fibroblasts (a normal human lung fibroblast cell line) showed that mitochondria can influence DNA damage response signaling and potentiate telomeric damage [63]. 

A correlation has also been observed between the degree of fibrosis in IPF and the aging of bone marrow stem cells (B-MSC). These B-MSCs have the potential to differentiate into AEC, and it has been shown that they express markers related to age. These cells characteristically are in the G0 phase of the cell cycle, which allows them to maintain their stability and their ability to initiate proliferation. However, this continuous resting status means that the cells do not activate DNA repair processes to eliminate harmful genetic alterations. Over time, these unrepaired mutations can accumulate, causing alterations in both the B-MSCs and the cells derived from them. A study in a murine animal model supports this hypothesis. The B-MSCs of older rats had no capacity for differentiation and proliferation, but the rats recovered when they were administered B-MSC from younger rats. This reinforces the idea that the development of pulmonary fibrosis is more frequent in people who have been smokers or have been exposed to other substances that damage the alveolar epithelium.

## 9. Mitochondrial Dysfunction

Mitochondrial function has an important impact on the aging process. Mitochondria not only play an exclusive role in providing energy to the cell in the form of ATP through oxidative phosphorylation processes, they are also part of a complex bidirectional intracellular signaling system between the cell nucleus and other organelles, and are partly responsible for maintaining intracellular homeostasis. Accumulation of dysfunctional mitochondria may be an important contributor to the pro-inflammatory aspects of cellular senescence. Mitophagy, the selective degradation of defective mitochondria by autophagy, is reduced in senescent cells [64].

Dysfunctional mitochondria can contribute to aging independently of ROS, as exemplified by studies in mice showing that increased levels of mitochondrial DNA (mtDNA) mutations result in accelerated signs of aging. It has been shown that mitochondrial dysfunction induced by mtDNA depletion induces senescence with a distinct phenotype, termed MiDAS (mitochondrial dysfunction-associated senescence) [65]. Aging and ER stress cause mitochondrial dysfunction in type II AEC by diminishing the expression of the mitochondrial homeostasis regulator PTEN-Induced Putative Kinase 1 (PINK1) [66,67]. Deficiency of PINK1 causes mitochondrial dysfunction characterized by alterations in mtDNA metabolism, and insufficient mitophagy, leading to increased susceptibility to apoptosis and induction of TGF-β, with increased susceptibility to lung injury and subsequent fibrosis. Other studies focusing on fibroblasts showed that TGF-β stimulation of lung fibroblasts decreased PINK1 levels and promoted insufficient mitophagy and myofibroblast differentiation.

These dysfunctional mitochondria are less efficient in antioxidant processing in AEC of IPF patients. This produces a specific profile of proinflammatory and pro-fibrotic cytokines with an increase in senescent fibroblasts that induces an increase in the enzymes that generate ROS. This redox imbalance plays an important role in maintaining a senescent phenotypic profile. Dysfunctional mitochondria activate multiple forms of damage-associated molecular patterns (DAMPS), such as ATP and mtDNA, but how the mitochondria regulate the SASP remains to be fully elucidated. Senescent cells show enhanced glycolytic activity, with relatively less ATP being generated by oxidative phosphorylation. AMP-activated protein kinase (AMPK) is a regulator of cellular responses to energy stress and is activated by increased AMP and ADP. AMPK activation can also lead to phosphorylation and activation of p53, promoting cell-cycle arrest via transcriptional upregulation of p21. Alterations in tricarboxylic acid (TCA) cycle activity have also been described in cellular senescence, however the extent to which this occurs is still not fully understood. Malate metabolism has also been implicated in senescence regulation, with loss of mitochondrial malic enzyme 2 (ME2) inducing senescence [68]. Several lines of treatment are being initiated to combat this type of mitochondrial dysfunction. It has been observed in vitro that restoration of the redox balance produces a decrease in the number of senescent cells and a reduction in fibrosis [69,70]. Several in vitro and ex vivo studies of lung cells with IPF have shown improvements in fibrosis with antioxidant treatment, and with senolytic agents such as dasatinib [62]. This new path opens a door to new treatment options for FPI.

## 10. Conclusions and Future Perspectives

Aging is associated with a wide variety of biological changes. Although interstitial lung disease can affect both the young and the elderly, aging is a significant risk factor for pulmonary fibrosis. IPF is an interstitial lung disease that is specifically associated with advancing age. The development of IPF is the result of multiple interrelated pathogenic pathways, included epigenetic, transcriptional, post-transcriptional, metabolic, and environmental factors, in individuals who are susceptible due to aging or who are genetically predisposed. More in vivo studies are needed with gain- or loss-of-function animal models to provide causal evidence for these proposed markers of the aging process. New lines of research using sequencing technologies may have a special impact on aging research, by facilitating the evaluation of genetic and epigenetic changes accumulated by individual cells in an aging organism. 

Currently FDA-approved drugs (pirfenidone and nintedanib) act to slow the progression of the disease. Recent studies of models of cellular aging in IPF open new treatment pathways, such as gene therapy aimed at reducing the early shortening of telomeres. These new approaches, together with molecular analysis of the genome-environment interactions that modulate aging, will help to identify new drug targets. A better understanding of the complex pathogenesis of IPF will lead to development of new therapeutic targets to attenuate or reverse lung fibrosis.

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
