# Peer review of "Causes of Pulmonary Fibrosis in the Elderly"

_medsci, 2018, doi:10.3390/medsci6030058_

Round 1

Reviewer 1 Report

This manuscript “Causes of pulmonary fibrosis in the elderly” by Lopez et al summarize and review the last decades of accumulated knowledge linking age and idiopathic pulmonary fibrosis. This is a very pleasant and informative reading and a great addition to this journal. This is an excellent review on a key topic in IPF. The authors have done a nice job of reviewing the available data and concisely presenting it in this manuscript. However, the authors can add some small comments to round the manuscript.

General comments:

The authors should be more specific in the type of cell they are refereeing to in each paragraph or if the study they are reviewing involved mice or patients. It seems most of the data only refers to type II epithelial cells.

Specific comments:

Section under “Genetic and epigenetic changes”. The authors could complete their manuscript mentioning some of the genetic defect in FPF (ie RTEL mutations). Also, work from Thannickal and Sanders has been groundbreaking in the role of histone acetylation in IPF. This manuscript will benefit if some comment about their work can be added.

Section under “Cellular senescence”, “Mitochondrial dysfunction”. The authors could complete their manuscript. Those section seems vague in the light of the new stream of data in both fields.

Author Response

Reviewer 1:

First of all we want to thank you very much for your interesting comments that surely improve the quality of our paper. All of them are in red color in the manuscript.

General comments:

The authors should be more specific in the type of cell they are refereeing to in each paragraph or if the study they are reviewing involved mice or patients. It seems most of the data only refers to type II epithelial cells.

Throughout the text we have tried to pointed out wich type of cells we are talking about, and also if it is a human or murine study when appropriate. All corrections are in red color in the text.

Specific comments

Section underGenetic and epigenetic changes. The authors could complete their manuscript mentioning some of the genetic defect in FPF (ie RTEL mutations). Also, work from Thannickal and Sanders has been groundbreaking in the role of histone acetylation in IPF. This manuscript will benefit if some comment about their work can be added.

Section underCellular senescence, Mitochondrial dysfunction. The authors could complete their manuscript. Those section seems vague in the light of the new stream of data in both fields.

We appreciate the comments of the reviewer. In section “Genetic and epigenetic changes” we have introduced some references about the role of histone acetylation in IPF and genetic defect in FPF. We notice about the importance of Thannickal work in this field. There are new comments from the Thannickal and Sanders works that you could see in red color. The new references are number 28, 30, 31, 32 and 35.

In the Section under “Cellular senescence” and “Mitochondrial dysfunction” we agree with the reviewer that we could complete our manuscript with update data. We have tried to improve them by adding some new data from recent publications in the field. We have revised all publications until June 2018 and we have added some paragraphs that you could see in red color, with updated references 53, 54, 55, 56 ,58, 59, 60,64 and 65.

Reviewer 2 Report

Comments to the manuscript/review entitled "Causes of Pulmonary Fibrosis in the Elderly by Lopez RC and coworkers

The very interesting review by Lopez RC and coworkers describes the role of the aging process in the pathogenesis of idiopathic pulmonary fibrosis. The aging process affects and impairs many cellular pathways, including telomere- and mitochondria-related maintenance-mechanisms, and proteostasis. I have a few comments, which should be considered.

Major comments:

1. COPD is also an age-related disease, but is "opposite" to IPF, because it is characterized by reduction of lung tissue, whereas IPF is characterized by irreversible scar tissue generation and generation of abnormal bronchiolar epithelium.

The authors should describe why aging can lead on the one side to emphysema-development and collagen-reduction and on the other side to fibrotic tissue generation. Which factors determine the fate of the aged lung to develop either COPD or lung fibrosis? For sure, smoking is a risk factor for developing IPF – but smoking is also the necessary second hit for COPD. Is the aged COPD-fibroblast more weak than the IPF-fibroblast?

Can the authors comment on this by creating an additional paragraph?

2. lines 60-62: "Radiographic studies have found alterations suggestive of interstitial disease when high resolution computed tomography was performed for unrelated reasons in asymptomatic elderly individuals with normal pulmonary function.“

Caution! In healthy lungs of aged individuals (>60 years old), thin alveolar walls and emphysema-like changes are predominantly present (own observations).

In general, the authors should describe more detailed the phenotype of the aged "healthy" human lung, with use of more literature (related to COPD/emphysema or interstitial lung disease?)

3. lines 70-71: The authors state: "These cells proliferate to replace damaged cells, sometimes exacerbating damage that might otherwise be asymptomatic."

The authors should explain this sentence more detailed, by writing additional three sentences or more.

4. lines 122 and 123: histone post-translational modifications: The authors should describe more detailed the epigenetic post-translational histone modifications (histone methylation, histone acetylation/deacetylation).

5. lines 107 and 108: These effects are mediated by increased endoplasmic reticulum (ER) stress….The authors should give at least one reference for this sentence.

6. line 126: In the elderly (>65 years), there is a progressive loss of DNA methylation…This reviewer wants to know in which paper this is stated.

7. lines 132 and 133: the authors state that reduced histone H3 and H4 acetylation in IPF-fibroblasts is associated with decreased COX-2 expression. This is correct. But does occur decreased histone acetylation = increased histone deacetylation during the aging process? This does not seem to be the case because COPD-lungs are characterized by a marked lack of histone deacetylases, thereby revealing abnormally increased acetylation status of histones. And COPD is considered as an age-associated disease. The authors should comment on this.

8. lines 120-151: The epigenetic part including the microRNAs should be given more detailed, and more references are needed.

Minor comments:

1. line 229: the activity of stem cells, as well as its replication….please write: as well as their replication

2. line 292: …pro-fibrotic cytosines….please write cytokines

3. Which are the references [45] and [46]?

Author Response

Reviewer 2:

Thank you very much for your interesting comments. We agree with all of them and we have tried to answer and complete the manuscript by adding a few paragraphs and updated references. Due to the addition of new paragraphs, the line numbers could have been changed in the new version. For explanation we use the lines from the first drafting. All of them are in red color in the new version, included references.

Major comments:

1. COPD is also an age-related disease, but is opposite to IPF, because it is characterized by reduction of lung tissue, whereas IPF is characterized by irreversible scar tissue generation and generation of abnormal bronchiolar epithelium.

The authors should describe why aging can lead on the one side to emphysema-development and collagen-reduction and on the other side to fibrotic tissue generation. Which factors determine the fate of the aged lung to develop either COPD or lung fibrosis? For sure, smoking is a risk factor for developing IPF – but smoking is also the necessary second hit for COPD. Is the aged COPD-fibroblast more weak than the IPF-fibroblast?

Can the authors comment on this by creating an additional paragraph?

We have created an additional paragraph about COPD and IPF. We have tried to explain why aging can lead on the one side to emphysema-development and on the other side to IPF. Some questions are very hard to answer because there are not  evidences enough.  Update references 8 and 11.

2. lines 60-62: Radiographic studies have found alterations suggestive of interstitial disease when high resolution computed tomography was performed for unrelated reasons in asymptomatic elderly individuals with normal pulmonary function.“

Caution! In healthy lungs of aged individuals (>60 years old), thin alveolar walls and emphysema-like changes are predominantly present (own observations).

 In general, the authors should describe more detailed the phenotype of the aged healthy human lung, with use of more literature (related to COPD/emphysema or interstitial lung disease?)

 We describe in more detail the phenotype of the aged  “healthy”  human lung and “senile” lung and,as the reviewer  said, in older patients there are emphysema-like and Interstitial lung abnormalities (ILA)  aged-related. Updated references 8, 11 and 13.

3. lines 70-71: The authors state: These cells proliferate to replace damaged cells, sometimes exacerbating damage that might otherwise be asymptomatic.

The authors should explain this sentence more detailed, by writing additional three sentences or more.

There is a new paragraph which tries to explain this sentence in more detail. Updated references 13 and 14.

4. lines 122 and 123: histone post-translational modifications: The authors should describe more detailed the epigenetic post-translational histone modifications (histone methylation, histone acetylation/deacetylation

Histone post-translational modifications. We have introduced some lines from important previous work in this field and we have added references about the role of histone acetylation in IPF and genetic defect in FPF from Thannickal and Sanders work. Updated references 26, 28 and 30.

5. lines 107 and 108: These effects are mediated by increased endoplasmic reticulum (ER) stress….The authors should give at least one reference for this sentence.

 We have tried to clarify this by updated  reference 23.

6. line 126: In the elderly (>65 years), there is a progressive loss of DNA methylation…This reviewer wants to know in which paper this is stated.

We have tried to clarify this by updated reference 26.

7. lines 132 and 133: the authors state that reduced histone H3 and H4 acetylation in IPF-fibroblasts is associated with decreased COX-2 expression. This is correct. But does occur decreased histone acetylation = increased histone deacetylation during the aging process? This does not seem to be the case because COPD-lungs are characterized by a marked lack of histone deacetylases, thereby revealing abnormally increased acetylation status of histones. And COPD is considered as an age-associated disease. The authors should comment on this.

We have tried to clarify this point in lines 193-204 of the new version, and we updated references 28, 29, 30, 31 and 32.

8. lines 120-151: The epigenetic part including the microRNAs should be given more detailed, and more references are needed.

We have added a new paragraph (new lines 210-224) and updated references 33, 34, and 35.

Minor comments:

1.       line 229: the activity of stem cells, as well as its replication….please write: as well as their replication. 

It has been updated

2.       line 292: …pro-fibrotic cytosines….please write cytokines

It has been updated

3. Which are the references [45] and [46]?

 It has been updated. Reference 45 is now reference 63 in the new version. Reference 46 was a  duplicate of reference 42 in the first draft and is now reference 57 in the new version.

Round 2

Reviewer 1 Report

This manuscript “Causes of pulmonary fibrosis in the elderly” by Lopez et al summarize and review the last decades of accumulated knowledge linking age and idiopathic pulmonary fibrosis. The manuscript has been revised satisfactorily however, it can benefit from some small additions.

Specific comments:

Section under “Environmental and endogenous factors”.

(1) The authors imply that telomere shortening is a consequence of the aberrant activation of AEC. There is a good body of literature that states the relationship inversely: telomere shortening can be the cause of a fragile epithelium. Please revise.

(2) The references used to back the statement “Results of human studies and animal models have shown an association 104 between IPF and Epstein-Barr virus (EBV) infection or g-herpes virus infection [18,19]” are too broad. Reference 18 is a new review on all types of fibrosis animal models (infectious and non-infectious, murine and non-murine) and reference 19 is a case study in a 27- years old. Probably any epidemiology study linking EBV infection and IPF will be more fitting.

(3) There is no reference to back the statement ”Studies in murine models have confirmed that aged wild- type mice experience more severe progressive pulmonary fibrosis than younger mice after EBV infection”. To my knowledge, EBV models are use in horses and not mice. They are several studies linking MHV68 and severity of fibrosis in older mice (work by Betty Moore and/or Ana Mora).

Section under “Loss of proteostasis”. Induction of ER stress in the AECII is a key piece in the pathobiology of IPF. The author tiptoe over it. The manuscript will benefit from an expanded revision of the role of ER stress in lung fibrosis.

Section under “Cellular senescence”.

(1) In the paragraph “IPF lung tissue and human IPF cells show increased senescence propensities ex vivo….” specific cell types are to be named. As it reads, it seem that the study referenced in [58] is done in AEC when it is not. Please revise.

Author Response

Thank you very much for your interesting comments . We agree with the comments of the reviewer and we have tried to answer all of them They are in color in the revised manuscript.

This manuscript “Causes of pulmonary fibrosis in the elderly” by Lopez et al summarize and review the last decades of accumulated knowledge linking age and idiopathic pulmonary fibrosis. The manuscript has been revised satisfactorily however, it can benefit from some small additions.

Specific comments:

Section under “Environmental and endogenous factors”.

The authors imply that telomere shortening is a consequence of the aberrant activation of AEC. There is a good body of literature that states the relationship inversely: telomere shortening can be the cause of a fragile epithelium. Please revise.

We have created an additional paragraph about the relationship between telomere and AEC and we have tried to better explain these not fully understood mechanism. As it’s being related to the telomeres we have included this in the  Telomere attrition  section.

“There is an association between numerous genes in the telomerase maintenance pathway and IPF but the mechanisms by which telomere defects provoke lung disease are not fully understood. There  are pathogenic  telomerase variants  causing dysfunction of telomerase activity leading to accelerated telomere shortening. In the other hand alveolar epithelial cells are necessary for repair after pulmonary injury after its activation an proliferation. Every replication of AEC DNA results in the decrease of telomere length . During successive cycles of cell division, telomere shortening occurs and eventually leads to activation of the DNA-damage pathways, which result in apoptosis or senescence”.

(2) The references used to back the statement “Results of human studies and animal models have shown an association 104 between IPF and Epstein-Barr virus (EBV) infection or g-herpes virus infection [18,19]” are too broad. Reference 18 is a new review on all types of fibrosis animal models (infectious and non-infectious, murine and non-murine) and reference 19 is a case study in a 27- years old. Probably any epidemiology study linking EBV infection and IPF will be more fitting.

We agree with the reviewer  and we have tried to clarify this by updated  with references [18,19,20,21,22]. We have added a new paragraph summarizing data from studies linking EBV infection and IPF. We have changed EBV for Gamma herpesvirus, that seems more appropriate. EBV is a gammaherpesvirus, so this could be confusing. There are studies with murine gammaherpesvirus, but not EBV exactly.

(3) There is no reference to back the statement ”Studies in murine models have confirmed that aged wild- type mice experience more severe progressive pulmonary fibrosis than younger mice after EBV infection”. To my knowledge, EBV models are use in horses and not mice. They are several studies linking MHV68 and severity of fibrosis in older mice (work by Betty Moore and/or Ana Mora).

We agree with the reviewer. EBV models are in horses. In mice models there are studies with  murine gammaherpesvirus.  We have updated this point with a new paragraph.

Section under “Loss of proteostasis”. Induction of ER stress in the AECII is a key piece in the pathobiology of IPF. The author tiptoe over it. The manuscript will benefit from an expanded revision of the role of ER stress in lung fibrosis.

We agree and we have updated this point with a new paragraph and 3 references. We have mentioned a recent review from Burman et al about this topic.

 Initial observations linking ER stress and IPF were made in cases of familial interstitial pneumonia (FIP), the familial form of IPF, in a family with a mutation in surfactant protein C (SFTPC). Subsequent studies involving lung biopsy specimens showed that  ER stress was implicated in the pathogenesis of IPF with the discovery of disease-causing mutations in surfactant protein C, which result in a misfolded gene product in type II alveolar epithelial cells (AECs) ( Lawson WE, Crossno PF, Polosukhin VV, Roldan J, Cheng DS, Lane KB, Blackwell TR, Xu C, Markin C, Ware LB, Miller GG, Loyd JE, Blackwell TS. Endoplasmic reticulum stress in alveolar epithelial cells is prominent in IPF: association with altered surfactant protein processing and herpesvirus infection.Am J Physiol Lung Cell Mol Physiol. 2008 Jun;294(6):L1119-26. doi: 10.1152/ajplung.00382.2007). ER stress and the unfolded protein response (UPR) have been linked to lung fibrosis through regulation of AEC apoptosis, epithelial-mesenchymal transition, myofibroblast differentiation, and M2 macrophage polarization (Tanjore H, Blackwell TS, Lawson WE.Emerging evidence for endoplasmic reticulum stress in the pathogenesis of idiopathic pulmonary fibrosis.Am J Physiol Lung Cell Mol Physiol. 2012 Apr 15;302(8):L721-9. doi: 10.1152/ajplung.00410.2011) . Although there are advances in knowledge, our understanding of the relationship between ER stress and lung fibrosis is lacking. Burman et al published recently a intersting review discussing potential causes of ER stress induction in the lungs and current evidence linking ER stress to fibrosis in the context of individual cell types: AECs, fibroblasts, and macrophages.  (Burman A, Tanjore H, Blackwell TS. Endoplasmic reticulum stress in pulmonary fibrosis.Matrix Biol. 2018 Aug;68-69:355-365. doi: 10.1016/j.matbio.2018.03.015)

Section under “Cellular senescence”.

         In the paragraph “IPF lung tissue and human IPF cells show increased senescence propensities ex vivo….” specific cell types are to be named. As it reads, it seem that the study referenced in [58] is done in AEC when it is not. Please revise.

We agree with the reviewer, this a  study in human MRC5 fibroblasts . It has been update. 

Reviewer 2 Report

The authors have revised their manuscript appropriately.

Author Response

N/A